# NMR analysis of the correlation of metabolic changes in blood and cerebrospinal fluid in Alzheimer model male and female mice

Filip Stojanovic[1], Mariam Taktek[1], Nam Huan Khieu[2], Junzhou Huang[2], Susan Jiang[2], Kerry Rennie[2], Balu Chakravarthy[2], Will J. Costain[2], Miroslava Cuperlovic-Culf[1]*

1 National Research Council of Canada, Digital Technologies Research Centre, Ottawa, Canada, 2 National Research Council of Canada, Human Health Therapeutics Research Centre, Ottawa, Canada

* miroslava.cuperlovic-culf@nrc-cnrc.gc.ca

**Data Availability Statement:** Data are available from the following: CSF metabolic profiles: NOESY experiment: https://nrc-digital-repository.canada.ca/eng/view/object/?id=1e764791-2e8d-46be-

## Abstract

The development of effective therapies as well as early, molecular diagnosis of Alzheimer's disease is impeded by the lack of understanding of the underlying pathological mechanisms. Metabolomics studies of body fluids as well as brain tissues have shown major changes in metabolic profiles of Alzheimer's patients. However, with analysis performed at the late stages of the disease it is not possible to distinguish causes and consequence. The mouse model APP/PS1 expresses a mutant amyloid precursor protein resulting in early Amyloid β (Aβ) accumulation as well as many resulting physiological changes including changes in metabolic profile and metabolism. Analysis of metabolic profile of cerebrospinal fluid (CSF) and blood of APP/PS1 mouse model can provide information about metabolic changes in these body fluids caused by Aβ accumulation. Using our novel method for analysis of correlation and mathematical ranking of significant correlations between metabolites in CSF and blood, we have explored changes in metabolite correlation and connectedness in APP/PS1 and wild type mice. Metabolites concentration and correlation changes in CSF, blood and across the blood brain barrier determined in this work are affected by the production of amyloid plaque. Metabolite changes observed in the APP/PS1 mouse model are the response to the mutation causing plaque formation, not the cause for the plaque suggesting that they are less relevant in the context of early treatment and prevention then the metabolic changes observed only in humans.

## 1. Introduction

Alzheimer's disease (AD) is an age-related, degenerative brain disease characterized by progressive cognitive impairment and dementia with increasing deficiency in multiple cognitive domains including memory, executive functions and language [1]. Pathological hallmarks of AD are neurofibrillary tangles composed of abnormally phosphorylated, conformed and truncated tau, and senile plaques with a core of the altered cleavage of amyloid precursor protein leading to the deposition of β-amyloid. In addition several other anomalies have been observed

a4fe-7d160ffb36b1 (DOI: https://doi.org/10.4224/40002051) Blood metabolic profiles: CPMG experiment:https://nrc-digital-repository.canada.ca/eng/view/object/?id=0ace4911-a04b-40d9-8d4a-7933df6b064b and this information has been added to the manuscript. (DOI: https://doi.org/10.4224/40002050).

**Funding:** The author(s) received no specific funding for this work.

**Competing interests:** The authors have declared that no competing interests exist.

including mitochondrial malfunction, increased oxidative stress and oxidative and nitrosative damage to nucleic acids, proteins and lipids; energy metabolism changes; altered composition of lipids and lipid rafts; autophagy; neuroinflammation, deregulation of purine metabolism and number of other metabolism pathways, dysregulation of unsaturated fatty acid metabolism, suggesting that metabolic changes are another hallmark of AD [2–7]. With predictions for a major increase in the number of AD patient's worldwide and associated major impact in human cost and suffering as well as economic impact there is a pressing need for better understanding of the AD etiology needed for the development of early diagnosis and effective treatment for this disease.

Metabolomics analysis provides information about the perturbations in the metabolome that reflect changes in genome, transcriptome and proteome caused by physiological as well as environmental impacts. Number of metabolomics studies have explored and determined significant metabolic profile changes in AD patients including measurements in cerebro-spinal-fluid (CSF), blood or brain tissue [8–10] with many reviews outlining results and problems [7,11,12]. A major obstacle in metabolomics studies of AD is the inability to discriminate metabolic changes that are resulting from brain pathology in AD from the metabolic changes that are preceding and thus possibly causing observed brain changes leading to AD development. Additionally, as AD is generally a disease of elderly, many different confounding health conditions are likely making determination of metabolic changes that are directly linked to AD very difficult. Transgenic amyloid precursor protein/presenilin 1 (APP/PS1) mouse model was created to mimic pathological and behavioral changes occurring as part of the AD [13]. The major goal and utilization of this transgenic model was aimed at testing and analysis of the drug and biomarkers of AD. An interesting alternative is to utilize this mouse model to determine changes in metabolic profile and metabolism that are caused by amyloid plaque formation. Understanding of metabolic changes that are promoted by APP/PS1 mutation would be valuable in subsequent determination of metabolic changes proceeding and possibly causing AD pathology. Metabolomics profiling of both cerebrospinal fluid (CSF) and blood allows analysis of correlation between metabolites in these two environments possibly indicating changes in the blood brain barrier function in the AD model. In this work we have explored metabolic changes in blood and CSF of male and female wild-type (WT) and APP/PS1 mice at two different age points and used different methods for machine and statistical learning as well as correlation analysis to describe sex-dependent effects of amyloid plaque on metabolic process.

## 2. Results and discussion

### 2.1. Qualitative NMR analysis

NMR metabolomics analysis of wild type (WT) and transgenic mouse model APP/PS1 was performed using 1D nuclear Overhauser effect spectroscopy (NOESY) pulse sequence for CSF and blood as well as Carr–Purcell–Meiboom–Gill (CPMG) experiment [14] for analysis of small molecules in the blood of mice as previously described [15]. In blood samples 1D NOESY measurements show combination of peaks for lipoproteins and small molecules. In CPMG measurement signals from the large molecules (lipids and proteins) are suppressed through relaxation, whereas the small molecules produce signal. This approach essentially makes large molecules invisible on 1H NMR without any filtration making measurement of both lipoproteins, with 1D NOESY and small molecule with CPMG, possible from one sample in quick succession [16].

Due to the small sample sizes and the relatively low sensitivity of NMR measurement, we have used a pool of samples from 3 mice for each measurement. Mice were sacrificed and CSF

and blood collected at early phase (2 months of age for both male and female mice) and mid-life point (7 months for male and 9 months for female mice). At 3 independent groups 3 mice were measured leading to total of 30 measurements representing samples from 90 mice. Details of the experimental procedure as well as NMR processing methods are outlined in Materials and Methods section. NMR spectra for CSF and Blood grouped by age and sex of mice are shown in Fig 1.

Principal component analysis (PCA) as well as t-distributed stochastic neighbor embedding (t-SNE) visualization transformation of complete spectra in three sample windows provides some separation of samples by APP/PS1 v.s. WT or sex. 1D NOESY spectra of blood samples show the most significant separation by APP/PS1 v.s. WT in PC1 and by sex in PC2 both at early and late time points (Fig 2). Further analysis of the loading shows that the major contribution to the PC1 sample separation is provided by peaks of lipoproteins, particularly very low density lipoprotein triglyceride–VLDL and saturated lipid peaks (Fig 2). It is important to point out that even before the appearance of visual plaques or deposits in the mouse model at 2 months of age there is a clear separation of samples provided by lipids and lipoprotein peaks in the blood with more significant separation by the disease model then by the sex. Recent analysis of a longitudinal cohort study in 318 patients presented by Nagga et al. [17] have shown that increased levels of triglycerides at midlife predict brain Aβ and tau pathology later in life in cognitively healthy individuals and that certain lipoprotein fractions present a risk factor for Aβ. Based on the results in mouse model shown here it can be suggested that initial

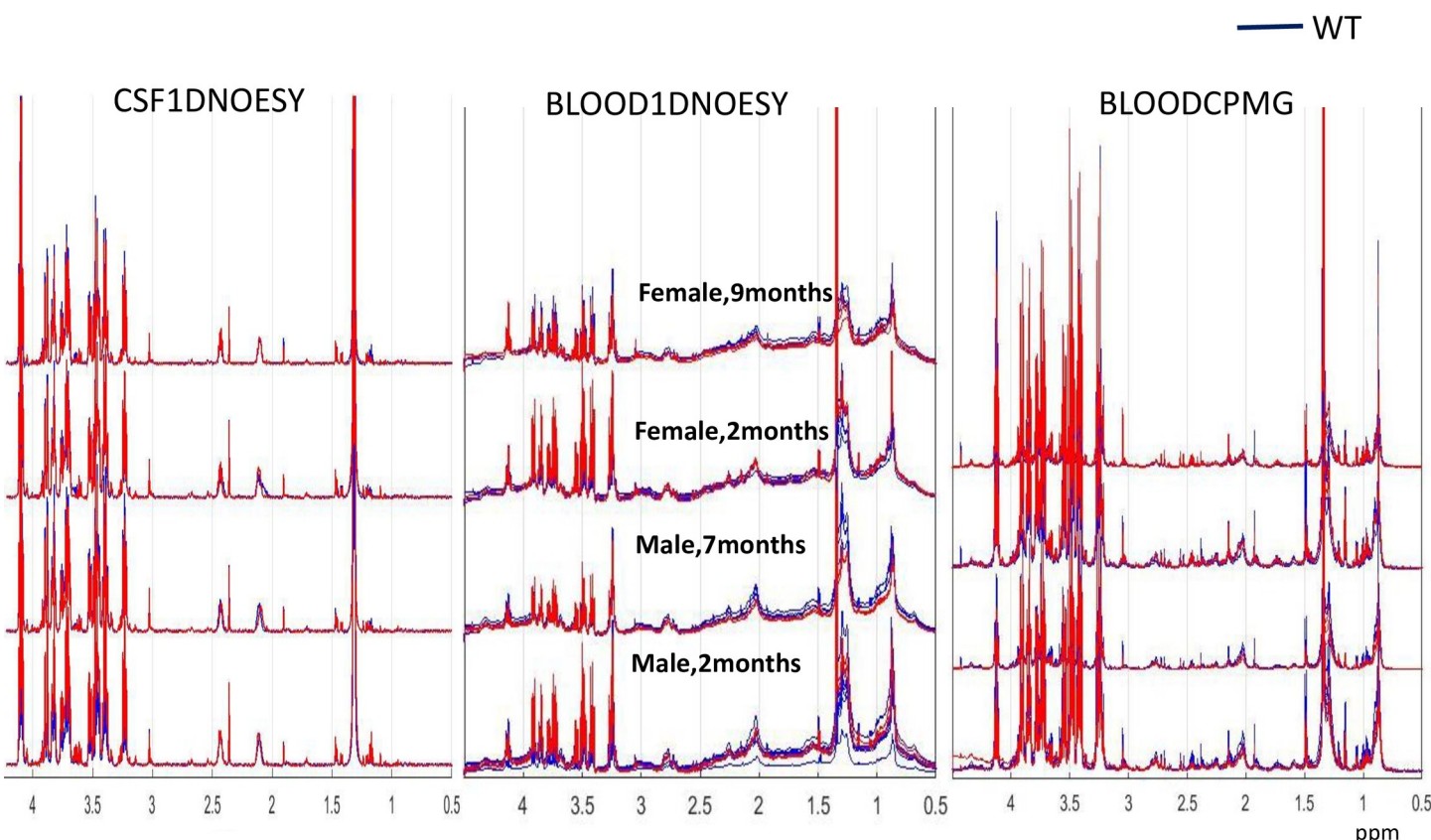

**Fig 1. NMR spectra for wild type (WT) and APP//PS1 mouse model (APP in the figure).** Each spectrum shows NMR measurement for the pool of 3 mice.

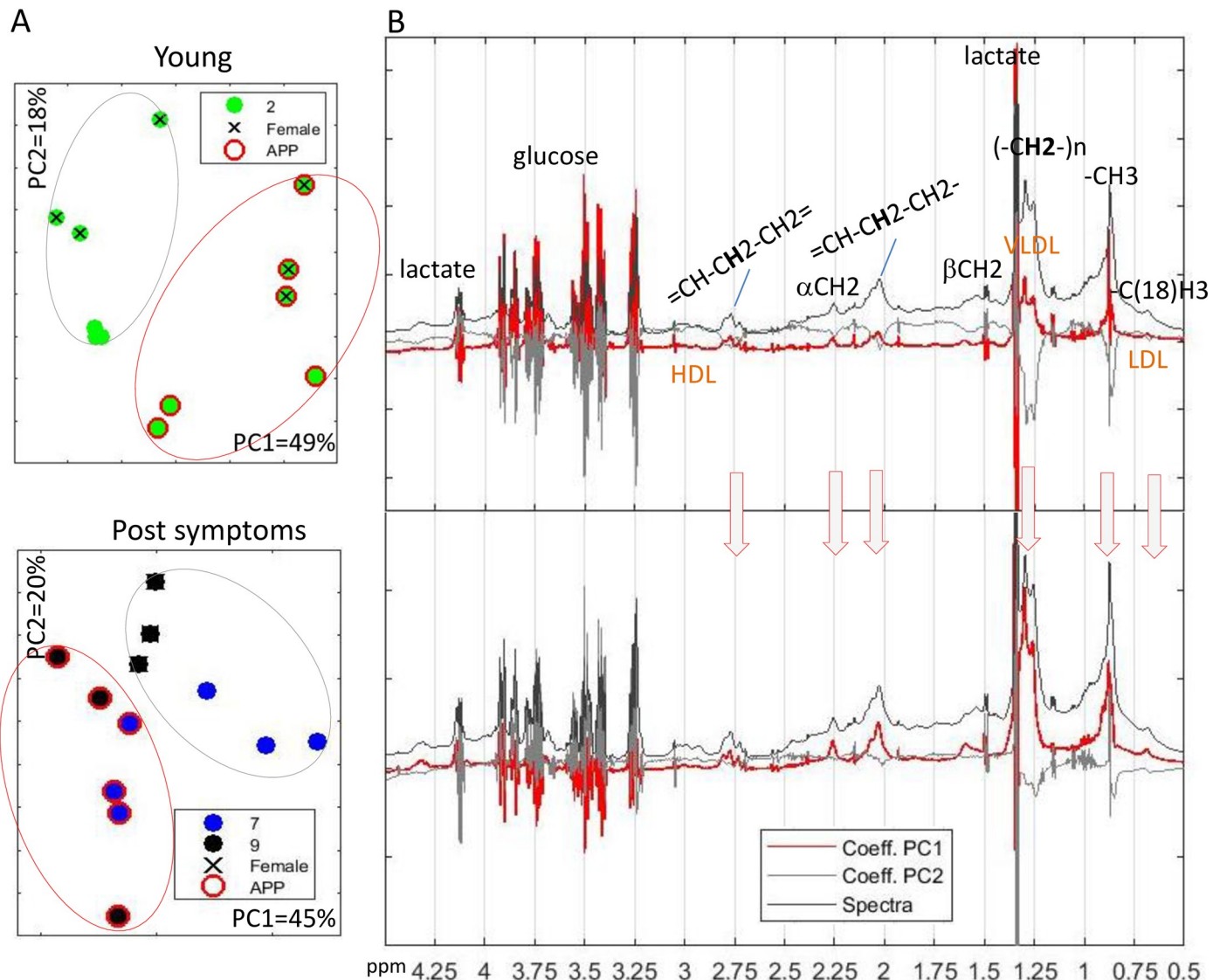

**Fig 2.** PCA of 1D NOESY spectra of blood showing sample separation (A) and loadings for PC1 and PC2 relative to the average spectra for all samples (B). Indicated are some representative peaks for cholesterol backbone (-C(18)H3), lipid (-CH3) as well as polyunsaturated fatty acid (= CH-CH2-CH2 =), fatty acids (= CH-CH2-CH2-) as well as possible peaks from lipoproteins of different sizes–low-density lipoprotein—LDL and high-density lipoprotein–HDL cholesterol and very low density lipoprotein triglyceride–VLDL. Figures shows different age groups (2, 7 and 9 months), mouse models with APP/PS1 (APP in figure) indicated with red circle and WT without and sex with female mice specified with x and male mice without.

Aβ formation even before any symptoms in fact leads to the observed changes in lipoprotein fractions as well as triglycerides, possibly suggesting that the observed lipoprotein changes in patients in the cohort are initiated by the initial Aβ formation process. Additionally, earlier work has shown that amyloid β-peptide is transported on lipoproteins and albumin [18,19] possibly explaining observed differences in lipophilic fraction early in the disease development.

The loading plot (Fig 2) shows that the significance of VLDL and LDL peaks on the PC1 observed separation of WT and APP/PS1 mice separation at later age point increases. This is in agreement with previously observed relationship between Aβ peptide and triglyceride-rich lipoproteins including VLDL and LDL [20] suggesting the possibility that the observed

differences in VLDL and LDL are the response to Aβ peptide and its clearance, in agreement with previous observations linking low-density lipoprotein receptor-related protein-1 (LRP1) and Aβ peptide clearance [21,22].

Unsupervised analysis of CSF spectral data (1D NOESY) and CPMG spectra of blood does not provide significant, unsupervised grouping of samples based on the disease model or sex (not shown).

## 2.2. Quantitative analysis and selection of major changes

In order to determine significant changes as well as possible relationships between metabolism and the disease development in APP/PS1 mice we have performed assignment and quantification of metabolites from NMR spectra using previously published methods [23–25] as briefly described in the Material and Methods. List of analyzed metabolites and their 1D standard spectra obtained from HMDB [26] or BMRB [27] are used for the determination of relative concentrations in the sample spectra are shown in Fig 3 with details of their selection and pre-processing described in Materials and Methods.

Quantification method determines the relative concentration for each metabolite independently of all the other metabolites using the constrained linear least-squares analysis and finding the best fit for the whole spectral range. By including the whole spectra, rather than only selected peaks, our methodology is able to provide relative quantification even for metabolites that have not fully resolved major peaks. The set of quantified metabolites can be divided into carbohydrate pathway metabolites, amino acids and derivatives, organic acids, and others. All quantified values for these metabolites, normalized only to the concentration of reference in CSF and blood in all groups of mice are available upon request. The change over time for each quantified metabolite in blood and CSF measurements is shown in Fig 4 as bar plot of the log of ratio of early v.s. late concentration value. It is apparent that changes over time in the blood and CSF in males and females are for number of metabolites very different. Therefore, finding a single panel of metabolic biomarkers that would provide high diagnostic sensitivity for different age groups and sex is highly unlikely for both blood and CSF even in these highly controlled animal models.

Based on the rate of change between two time points and in different sample types metabolites can be divided into several behavioral groups (Fig 4). Several metabolites show the same type of change in male and female mice in WT and APP/PS1 in either blood or CSF. Namely in blood there is a greater increase in APP/PS1 relative to WT is observed in both male and female mice for phenylalanine, isoleucine, taurine, homocysteine, acetone, acetate, o-phosphoethanolamine, threitol, myoinositol, galactose and pantothenic acid. The larger decrease in concentration over time in blood of APP/PS1 mice in both male and female groups is apparent for creatine. In CSF, the concentration in APP/PS1 decreases at a later time (to the point of more negative change) for ethanol, citrulline, serine, d-serine, valine, isoleucine, cysteine, gaba, homovanillic acid, citrate, malate, arabitol, lactate. On the other hand concentrations of ethanolamine, glucose 1-phosphate, glucose, are increasing at later time points faster in APP/PS1 male and female mice than in WT. Metabolites with statistically the most significant concentration difference between WT and APP/PS1 in both sexes at different ages are determined using ANOVA and are provided in Fig 5.

Previous work has shown change in metabolite profile over time in mouse model of AD [28,29] suggesting possible differences in metabolism, as well as possibly altered transport of metabolites across the blood brain barrier. Therefore, it is not surprising that metabolites showing the most significant concentration difference between APP/PS and WT at young and older age in blood and CSF are different. It is interesting to point out that several of the

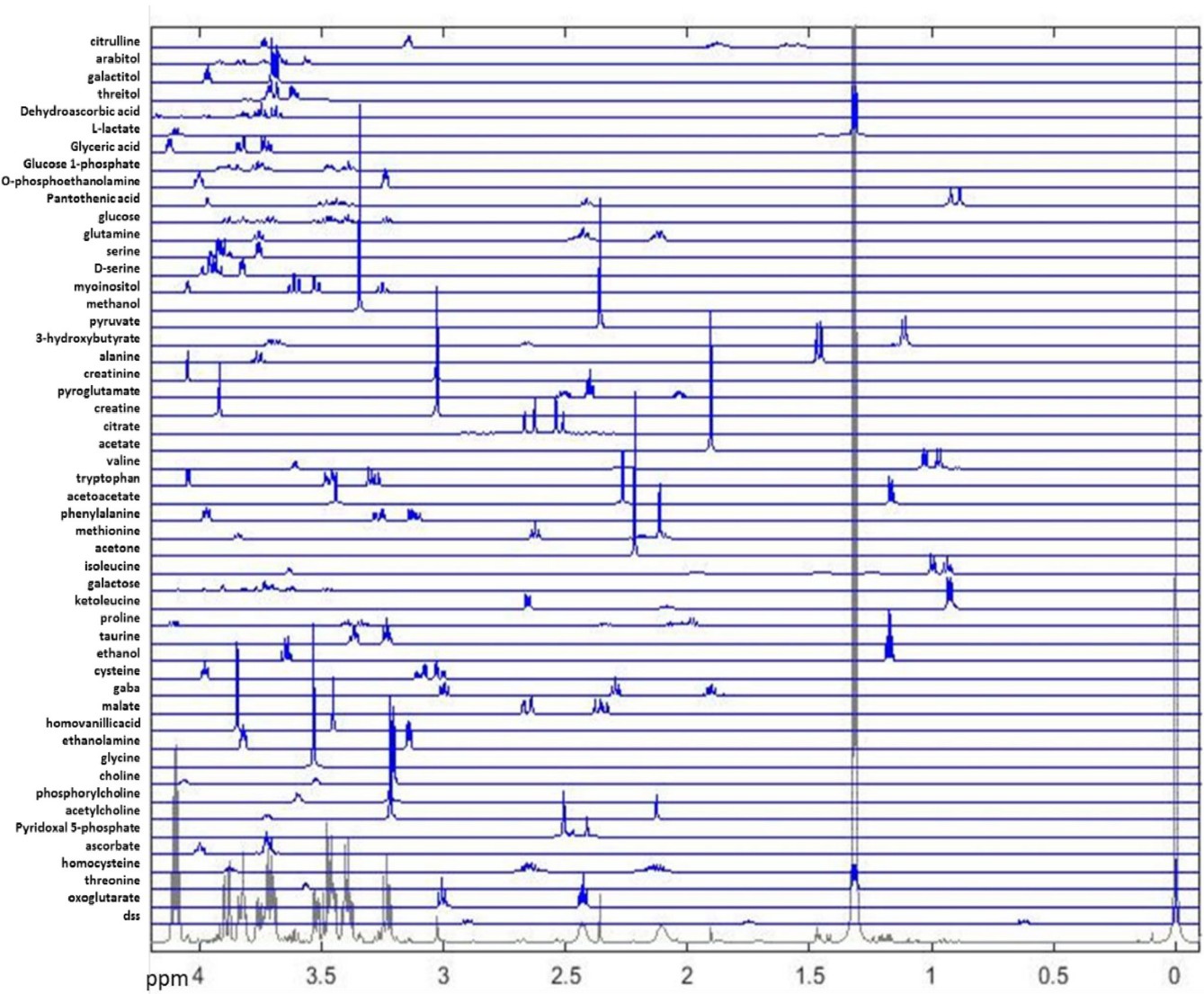

**Fig 3. Metabolites determined by ASICS [24] and Metabohunter [25] as possible observed in CSF and blood NMR.** Shown are complete spectral regions used for metabolite quantification.

metabolites shown as significant here have been previously observed as changed in AD both in patients and animal models. These results will be further explored together with correlation analysis presented below.

## 2.3. Correlation analysis of metabolites in CSF and blood

For shared metabolites, defined as metabolites present in both blood and CSF for a group of mice (male-WT; male-APP/PS1; female-WT; female-APP/PS1), Spearman correlation matrices were calculated between the concentrations of each shared metabolites in blood and each shared metabolite in CSF ($M$) and $v.v.$, each shared metabolite in CSF with each shared metabolite in blood ($M^T$). For $n$ shared metabolites $M$ is an $n$ x $n$ matrix where $M_{ij}$ corresponds to the correlation between the $j^{th}$ metabolites in CSF and the $i^{th}$ metabolites in blood. For each correlation, a significance level of $\alpha = 0.05$ was used to test the alternative hypothesis that the

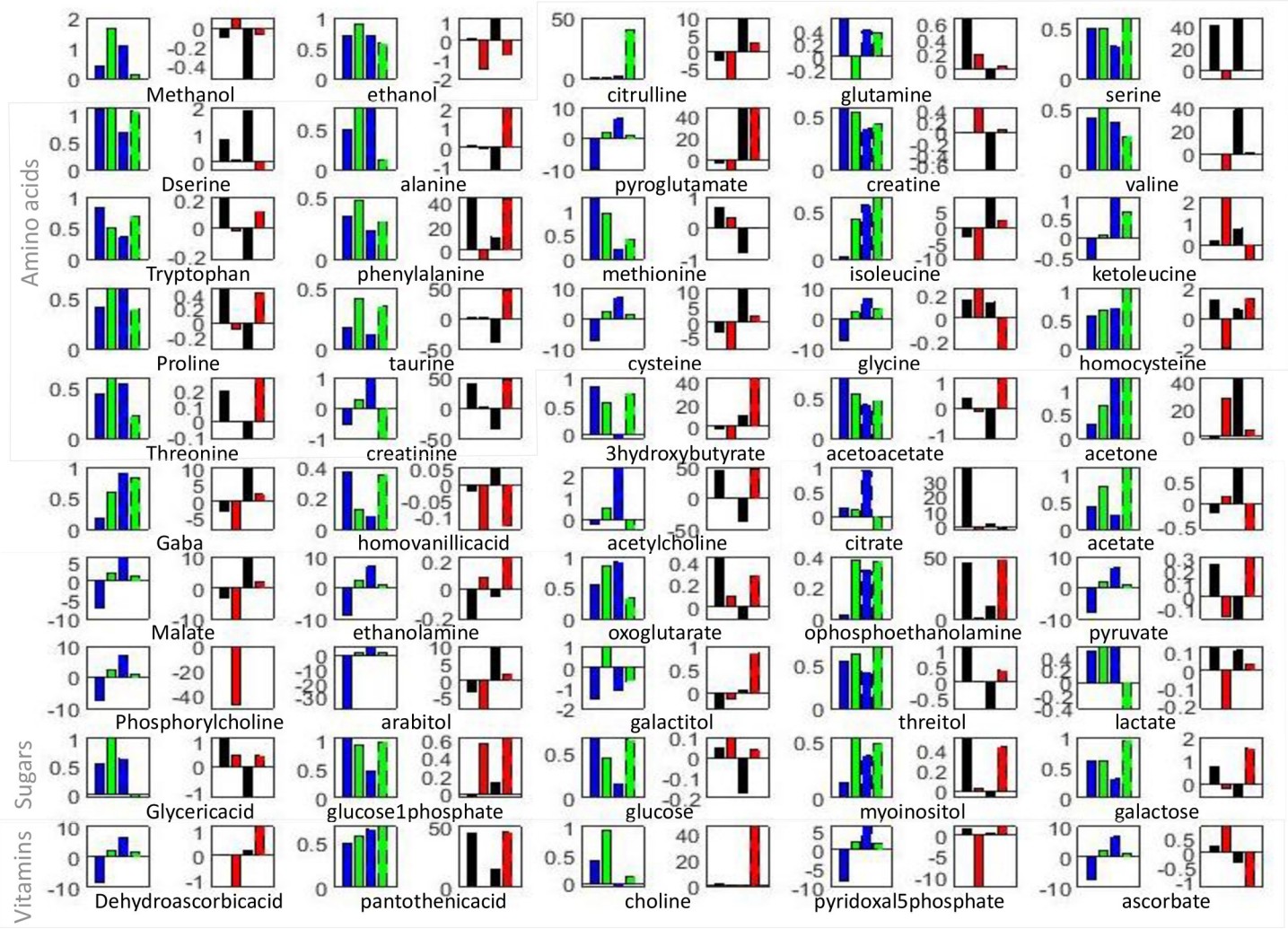

**Fig 4. Bar plot of the Log of the ratio between early and late time points metabolite concentrations in blood (WT blue/APP/PS1 green) and CSF (WT black/APP/PS1 red).** Female mice values are indicated with dashed lines (second pair on the graph).

correlation was different from 0 with $p$-values for the Spearman correlations computed as described in Materials and Methods. Metabolite pairs with statistically significant correlations were listed for each group of mice. If two metabolites were significantly correlated in one sex's WT mice, that same pair and its correlation in the APP/PS1 mice samples was added to the list of significant correlations for that sex's APP/PS1 mice, and *v.v.*, ensuring that both APP/PS1 and WT groups of each sex listed the same metabolite pairs for comparisons.

For each group of mice, an adjacency matrix $A$ listing correlation pairs was created, where $A_{ij}$ is the correlation between metabolite $i$ in blood and metabolite $j$ in CSF if the correlation is significant and 0 otherwise. We define a directed correlation network for each mouse group using its adjacency matrix, where each node represents a metabolite and each edge going from node $j$ to node $i$ represents a correlation between metabolite $i$ in blood and metabolite $j$ in CSF (and v.v. provided by $A^T$). To summarize the correlations that were statistically different from each other between the WT and APP/PS1 groups of each sex of mice, the correlation for each metabolite pair was transformed using the Fisher z-transformation. The difference between the transformed correlations of the same metabolite pair in APP/PS1 and WT mice was then

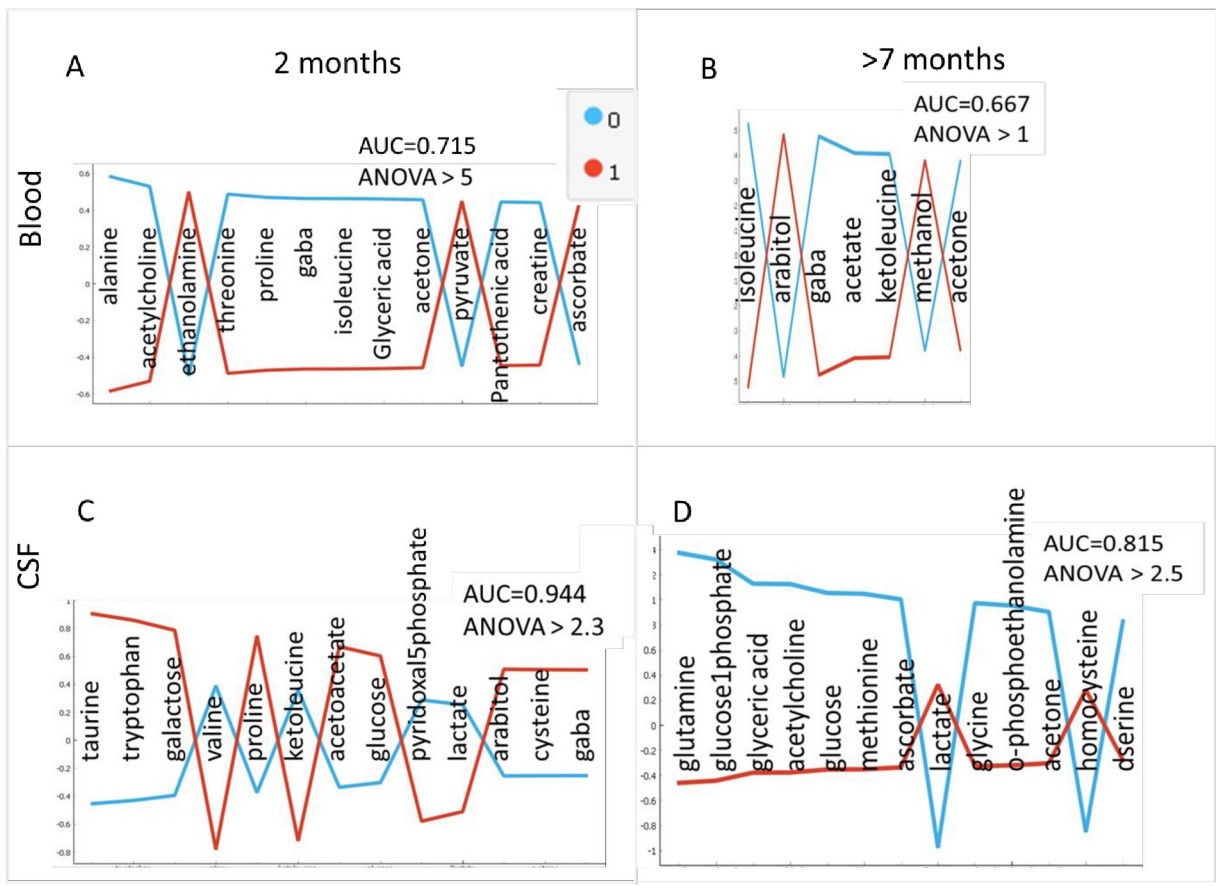

**Fig 5.** The most significantly differently concentrated metabolites in Blood (A and B) and CSF (C and D) between WT (blue line) and APP/PS1 (red line) subgroups including both male and female at 2 months old (A and C) and at later time points of 7 months for male and 9 months for female mice (B and D).

tested at significance level α = 0.05. CSF and blood metabolite partners with significantly different correlation levels between APP/PS1 and WT mice for male and female are shown in Table 1.

All significant correlations for 4 groups of animals between all metabolites in CSF and all metabolites in blood measured using CPMG is shown schematically in Fig 6.

## 2.4. Ranking of metabolites by correlation significance

We make use of a path counting algorithm to rank the metabolites in our network using newly developed method described in Materials and Methods. Our approach measures the connectedness of each metabolite with its column sum, providing ranking of connections for each metabolite. Since the first *n*-columns correspond to metabolites in blood and the last *n*-columns correspond to metabolites in CSF, we rank the metabolites in blood and CSF using the first *n* and the last *n* columns respectively. We show the differences in blood metabolite rankings between the APP/PS1 and WT mice of each sex in Fig 7.

Increase in the correlation ranking in APP/PS1 for both male and female mice is apparent in blood based acetate, acetone, gaba, glucose-1-phosphate, glyceric acid, isoleucine, o-phosphoetanolamine, serine and to a lesser extent 3-hydroxybutyrate suggesting increased traffic between CSF and blood of these metabolites or metabolites that are closely related with this

**Table 1. Significantly different correlations between female and male APP/PS1 (APP for brevity) and WT mice.**

| Female | | | | Male | | | |
|---|---|---|---|---|---|---|---|
| Blood | CSF | APP | WT | Blood | CSF | APP | WT |
| acetate | taurine | 0.89 | -0.43 | ethanol | acetate | -0.94 | -0.25 |
| galactose | ethanolamine | -0.66 | 0.74 | arabitol | oxoglutarate | 0.49 | -0.70 |
| homocysteine | ascorbate | 0.31 | -0.81 | creatine | creatine | 0.94 | 0.33 |
| methionine | valine | -0.49 | 0.81 | pantothenicacid | creatine | 0.94 | 0.30 |
| Pantothenic acid | ascorbate | 0.20 | -0.88 | 3hydroxybutyrate | lactate | 0.94 | -0.50 |
| 3hydroxybutyrate | myoinositol | 0.77 | -0.81 | Homovanillic acid | Homovanillic acid | 0.89 | -0.55 |
| isoleucine | acetylcholine | 0.94 | -0.05 | glutamine | oxoglutarate | -0.94 | 0.05 |
| acetylcholine | ethanolamine | 0.94 | -0.48 | | | | |
| acetylcholine | threitol | 0.66 | -0.79 | | | | |
| Glyceric acid | ethanol | 0.14 | -0.95 | | | | |
| Glyceric acid | ethanolamine | 0.94 | -0.26 | | | | |
| Glyceric acid | galactitol | 0.89 | -0.50 | | | | |
| citrulline | valine | -0.89 | 0.17 | | | | |
| glutamine | ethanolamine | -0.71 | 0.90 | | | | |
| glutamine | galactitol | -0.66 | 0.88 | | | | |
| glutamine | lactate | -0.26 | 0.90 | | | | |
| Homovanillic acid | glycine | 0.89 | -0.07 | | | | |
| choline | lactate | -0.89 | 0.43 | | | | |

Metabolites that show the most significant differences in concentrations between WT and APP/PS1 (from Fig 4) are underlined. Spearman's correlation coefficient for the pair of blood-CSF metabolites is indicated for both APP/PS1 model and WT mice.

group through metabolic processes. Decrease in correlation ranking in APP/PS1 in both male and female mice is observed for ethanol and glutamine. Number of other metabolites show different changes in connectivity in APP/PS1 between male and female mice. Positive and negative correlations between metabolites in CSF and blood in WT, APP/PS1, male and female mice is presented in Table 2. Within CSF based metabolites the most striking is the major reduction in connectedness for ascorbate in APP/PS1 in both male and female mice suggesting reduced transfer across the blood brain barrier in APP/PS1 mice.

Several of the metabolites with the statistically significant concentration difference between 2 month old, WT and APP/PS1 mice of both sexes in blood and CSF are shown to be highly correlated between blood and CSF. Pantothenic acid (vitamin B5) is needed to form coenzyme-A and is thus a critical metabolite in synthesize of carbohydrates, proteins and fats. Concentration of this metabolite in blood is higher in WT mice at 2 months than in APP/PS1 mice. At the same time its concentration change in blood is strongly correlated with number of CSF metabolites including glyceric acid, glucose-1-phosphate, pyruvate, creatine, galactose as well as ascorbate (vitamin C). Interestingly correlation between pantothenic acid and ascorbate is positive in APP/PS1 for both male and female mice but in female mice correlation between blood based pantothenate and CSF based ascorbate is negative (Fig 5).

Relative concentration of acetone in blood at 2 months is significantly lower in APP/PS1 mice and further decreases in this group of mice at a later time. In CSF, levels of acetone are very low in most cases except for APP/PS1 male mice at a later time point. Concentration change of acetone in blood is strongly correlated with number of CSF metabolites including glyceric acid, glucose-1-phosphate, pantothenic acid, creatine, galactose, proline, ketoleucine, taurine, acetylcholine, ascorbate and threonine.

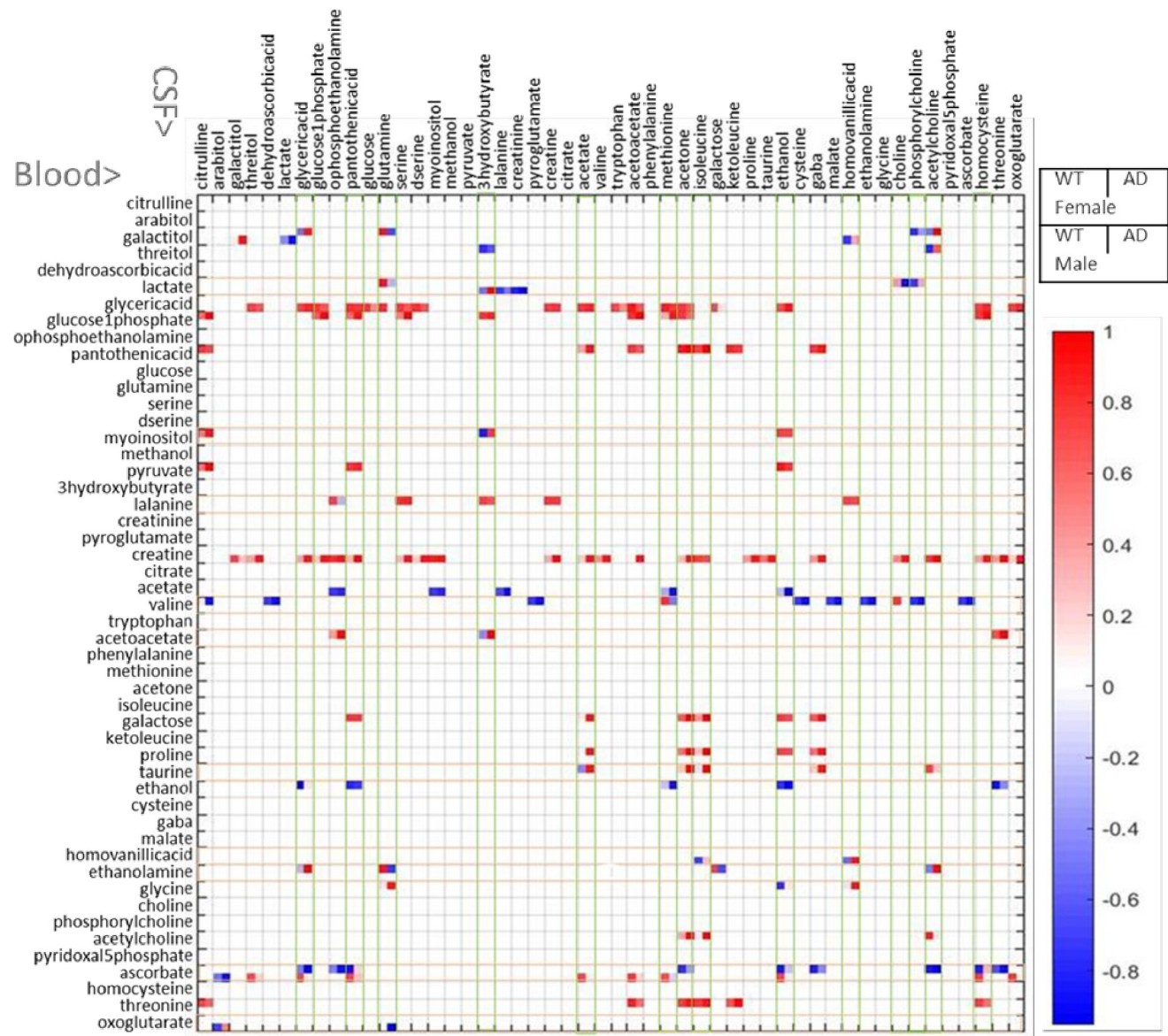

**Fig 6. Significant correlations between metabolites in blood (shown in column) and CSF (row) represented in 2x2 representations for WT and APP/ PS1 female and WT and APP/PS1 male.**

Although ascorbate (vitamin C) shows larger concentration in blood of AD mice it is important to keep in mind that its concentration in blood measured here is very low and thus possibly inaccurate. At the same time concentration of ascorbate in CSF is much higher and reduces at older APP/PS1 mice in both sexes suggesting its use in higher level of ROS environment in APP/PS1 mutants. Concentration change of ascorbate in CSF in highly correlated with number of blood metabolites including many metabolites with significant concentration difference in WT and AD mice blood including: glyceric acid, pantothenic acid, gaba, acetylcholine and threonine. Interestingly, correlation ranking for CSF ascorbate with blood metabolites in APP/PS1 mice is reduced possibly due to its utilization as antioxidant or reduced transport across the blood brain barrier in the disease model. Additionally, correlation between CSF ascorbate and blood based pantothenic acid and homocysteine goes from

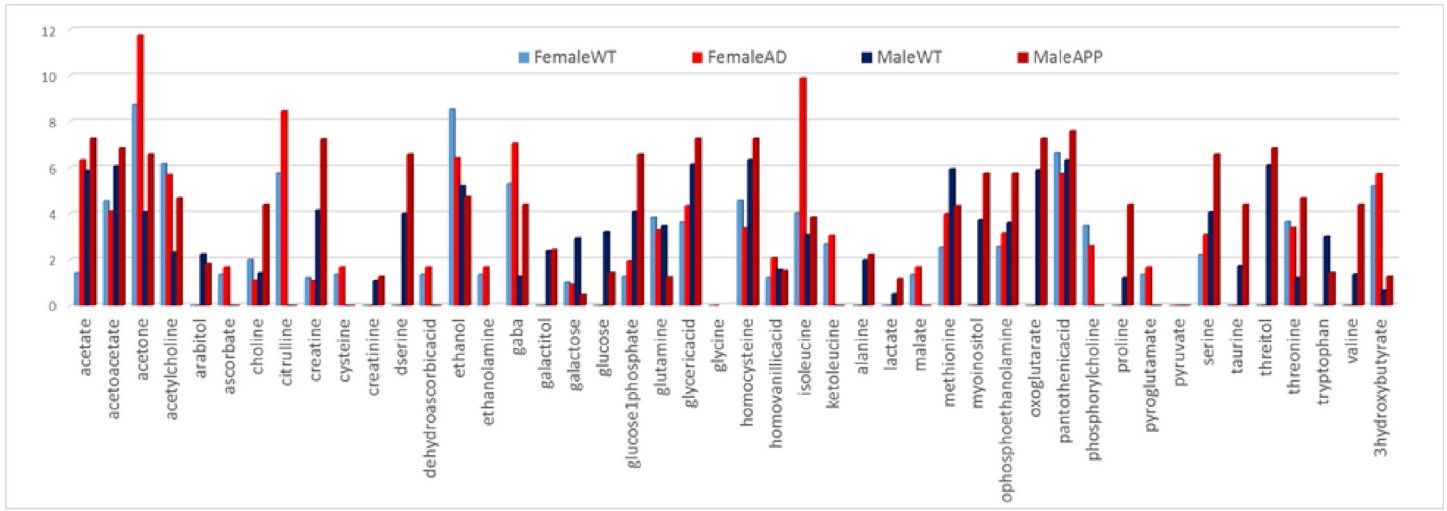

**Fig 7. Ranking of metabolite correlation between blood and CSF with showing ranking of blood based metabolites.**

negative to positive (Table 1). Relationship between homocysteine, as a measure of three B vitamins (folate, vitamin B12 and vitamin B6) and AD has been previously outlined [30] and our results show that the change in homocysteine as well as change in its relationship to ascorbate are result of amyloid plaque development. Furthermore, the relationship between different vitamins, including ascorbate, homocysteine and pantothenic acid are affected by the production of amyloid plaque, suggesting that their observed changes in AD are the response to plaque formation, not the cause for the plaque.

## 3. Conclusions

Comparison of the metabolic profiles of CSF and blood at different time points in male and female mice from the APP/PS1 mouse model has been used to investigate metabolic changes induced by the production of amyloid plaque. Even at an early, pre-symptomatic stage, there

**Table 2. Ranking of the connectedness of CSF metabolites with blood-based metabolites.**

| Female | | | | Male | | | |
|---|---|---|---|---|---|---|---|
| Metabolite | WT | Metabolite | APP | Metabolite | WT | Metabolite | APP |
| ascorbate | 10.6 | Glucose 1phosphate | 11.6 | Glyceric acid | 22.9 | creatine | 33.5 |
| Glucose 1phosphate | 8.8 | Pantothenic acid | 10 | creatine | 14.6 | Glyceric acid | 19.8 |
| valine | 8.2 | valine | 9.5 | ascorbate | 12.8 | acetate | 6.5 |
| Pantothenic acid | 7.9 | galactose | 9.1 | acetate | 4 | ascorbate | 3.3 |
| threonine | 6.9 | proline | 7.9 | lactate | 2.4 | lactate | 2.8 |
| galactose | 5.7 | ascorbate | 7.5 | Homovanillic acid | 1.7 | galactitol | 2.6 |
| ethanol | 5.6 | threonine | 7.2 | galactitol | 1.5 | oxoglutarate | 1.7 |
| alanine | 4.8 | taurine | 7.2 | oxoglutarate | 1 | Homovanillic acid | 1.4 |
| pyruvate | 3.9 | ethanol | 4.5 | | | | |
| galactitol | 3.8 | ethanolamine | 4.4 | | | | |
| acetoacetate | 2.1 | threitol | 1.9 | | | | |
| glycine | 1.7 | lactate | 1.8 | | | | |

Underlined are metabolites shown to have significantly different concentration in WT and APP/PS1 (APP in the table) in blood or CSF (Fig 4).

are several changes in lipoproteins and metabolites as well as changes in correlation between metabolites across the blood brain barrier. This analysis shows that several of the previously observed metabolic differences between AD and healthy subjects, such as differences in the lipoprotein levels, can be a result of amyloid plaque production rather than the cause for this hallmark of AD. The novel approach for the analysis of correlation between metabolites in the CSF and blood has shown changes in the number of significant correlations across the blood brain barrier for a number of metabolites indicated for their importance in AD, such as ascorbate, pantothenic acid, glucose 1-phosphate, glyceric acid and creatine. Change in the correlation level between metabolites in CSF and blood indicates change in the transport of some metabolites across the blood brain barrier. Elucidating the relationship between amyloid plaque formation and the observed metabolic changes is needed in order to confirm that the observed metabolic changes in this mouse model results from the amyloid plaque. Further analysis of metabolite correlation between CSF and blood in human AD patients is underway to explore changes resulting from amyloid plaque formation as well as possible metabolic precursors to the catastrophic changes in the brain.

## 4. Materials and methods

### Animals and sample collection

Double transgenic AD mice (B6.Cg-Tg) harboring PSEN1dE9 and APP$_{Swe}$ transgenes that accumulate oligomeric A$\beta_{1-42}$ aggregates and their corresponding wild type strain C57BL6 that do not were obtained from Jackson Laboratory and maintained at NRC. The animals were housed in groups of three in a 12 h light-dark cycle at a temperature of 24˚C, a relative humidity of 50 ± 5%, and were allowed free access to food and water. All animal procedures were approved by the NRC's Animal Care Committee (NRC Research Ethics Board Secretariat) and were in compliance with the Canadian Council of Animal Care guidelines.

Mice were anesthetized with isoflurane and placed in a stereotaxic instrument with the head rotated downward at a 45˚ angle. The body was covered with a pre-warmed blanket to maintain the body temperature. A midline incision was made with fine scissors between the ears beginning at the occipital crest and extending caudally about 1.5 cm on the back of the neck. The subcutaneous tissue and muscles covering the cisterna magna were separated by blunt dissection. The neck muscles were retracted in order to expose the dura mater. Under a microscope, the dura mater of the cisterna magna appeared as a glistening with a clear reverse triangle through which the medulla oblongata, cerebellum, blood vessels, and the CSF space are clearly visible. The dura mater was blotted dry with a fine cotton swab to facilitate the careful selection of a spot for penetration of a glass capillary while avoiding vessels, thereby preventing blood contamination. The pulled capillary tube (Sutter Instrument, BF120-94-10, pulled on a Sutter P-97 Flaming micropipette puller with a heat index of 300 and pressure index of 330) was carefully inserted into the cisterna magna through the dura mater. CSF flowed into the capillary tube following a noticeable change in resistance to insertion. The capillary tube was then carefully removed, and connected to a 1 mL syringe through a 27G butterfly needle tubing (about 5 cm). The CSF was then injected into a sample vial (Waters, Ca#186000384c), and the vial was immediately frozen on dry ice and stored at -80˚C until further analysis. Following CSF collection, blood (around 500 μL) was collected into a BD Microtainer tube (BD 365967) by heart puncture. Blood was left at room temperature for 15–30 min, and then centrifuge at 1500 rpm for 10 min. The serum was transferred to a sample vial, immediately frozen on dry ice and stored at -80˚C until further analysis. Euthanasia was performed by cervical dislocation under deep isoflurane anesthesia.

CSF samples were prepared for NMR by adding 16 μL 3-(Trimethylsilyl)-1-propanesulfonic acid-d6 sodium salt (DSS-D6; Chenomix/Sigma-Aldrich, Oakville, ON, Canada) to 30 μL CSF (pooled from 1–3 animals) and placing in 3 mm NMR tubes (Wilmad®, 7 inch; Sigma-Aldrich). Serum samples (samples pooled correspondingly to the CSF samples) were prepared for NMR by combining 16 μL standard with 50 μL serum and placing in a 3 mm NMR tubes (Wilmad®, 7 inch; Sigma-Aldrich). Buffer was not added to the samples and minor peak shifts due to small pH changes in the body fluids was corrected computationally.

## Experimental analysis

All $^1$H NMR measurements were performed on a Bruker 600 MHz spectrometer at 298 K. 1D $^1$H (proton) NMR were measured for all samples using 1D $^1$H with water suppression sequence. Blood samples were measured using CPMG and NOESY 1D pulse sequences provided by Bruker, while the CSF samples were only measured using NOESY 1D sequence. All spectra were processed using MestReNova 9.1.0 software (Mestrelab Research Solutions) with specific preprocessing steps including: exponential apodization (exp 1); global phase correction; and normalization using the reference peak at 0ppm. Spectral regions from 0.5–9.5 ppm were included in the normalization and analysis with water peak region (4.5–5.2ppm) removed. Spectra were saved as txt files and imported into Matlab (Matworks Inc) for subsequence processing and analysis. The average of the spectral intensity up-field from -3 ppm and downfield from 12.5 ppm was subtracted from the spectra as a simple baseline correction. Subsequently the points were binned by taking the average over sections with a width of 0.0008 ppm.

Metabolite assignment was performed using 1D $^1$H data in comparison to the standard metabolite spectra provided in databases HMDB [26] and BMRB [27] and NMR search tools MetaboHunter [25] and ASICS [24]. Additionally, all automated assignments have been confirmed based on literature information for metabolites' measurement in related samples. A total of 51 metabolites were included in the analyses of both CSF and blood. Standard $^1$H spectra for assigned metabolites were obtained from the Human Metabolomics Database (www.hmdb.ca) or Biological Magnetic Resonance Databank (www.bmrb.wisc.edu) and processed using the same procedure as above with MestReNova 9.1.0 software. Prior to quantification procedure, the metabolite standard spectra were aligned to the reference peak (trimethysilyl-propoinate) using peak alignment by fast Fourier transform cross-correlation [31].

A semi-automated method, running under Matlab, for quantification using multivariable linear regression has been developed and described previously [23]. Our approach finds the best fit for the complete spectral region for each selected metabolites to the measured 1D sample spectra. This partial least square regression analysis result was used as the starting point and the model was constrained to concentrations greater than or equal to zero and re-run using Levenberg-Marquardt curve fitting using lsqcurvefit method in Matlab. Metabolite concentrations across samples were determined using the same standard spectra that were normalized to the total intensity equal to one and sample spectra normalized to the reference. Therefore, regression analysis provides relative metabolite concentration measures in different samples. By using the same standard scale based on the reference peak obtained relative concentrations allow comparison between samples without the absolute metabolite concentrations.

## Data analysis

Pre-processing including data organization, removal of undesired areas, z-score normalization, as well as data presentation was performed with Matlab vR2010 and vR2017a

(Mathworks). Minor adjustments in peak positions (alignment) between different samples were performed using Icoshift [32]. Both linear, Principal Component Analysis (PCA) and non-linear t-Distributed Stochastic Neighbor Embedding (t-SNE) techniques running under Matlab have been used to visualize high-dimensional spectral and quantified metabolite data for all three metabolome windows. Principal component analysis (PCA) was performed in Matlab using routine ppca for probabilistic principal component analysis.

Machine learning and statistical methods running under Matlab and Orange, a component based data mining software running under Anaconda Python Data Science Platform (https://anaconda.org/; https://orange.biolab.si/) were used for feature selection in different groups. Specifically, feature selection was done using Logistic Ridge Regression performing L2 regularization as well as ANOVA ranking as presented in Orange and metabolites selected by both methods were chosen as significant.

## Data processing and analysis

**Analysis of correlation between metabolites in CSF and blood.** R routine utilizing ASICS R package was used to identify metabolites in the CSF and blood samples of each of the four groups of mice we studied including male, female; APP and WT mice at different age points. Correlation between metabolites measured in both blood and CSF was determined using Spearman method in R using AS89 algorithm. Spearman correlations were calculated between the concentrations of each shared metabolite identified in blood and each shared metabolite identified in CSF. Correlations between each metabolite in blood and each metabolite in CSF was considered if above significant level of 0.05. Statistically different correlations between metabolites in two sample groups (blood and CSF) between WT and AD groups in two sexes are determined by transforming the correlation for each metabolite pair using Fisher z-transformation and then the transformed correlation have been tested at significance level a = 0.05. The metabolite pairs that were identified as significantly different from each other in WT and AD groups are summarized in Table 1. Ranking of the correlation significance for each metabolite in each group (male, female, APP or WT) was done using ranking method developed in house and described below.

**Ranking of metabolites by correlation significance.** We make use of a path counting algorithm to rank the metabolites in our network. For this, we use the absolute value of the adjacency matrix because we are primarily interested in the ability of a metabolite to affect others, which is described by the magnitude of a correlation. Any two metabolites can influence each other directly as well as indirectly, i.e. through any number of ancillary metabolites. Therefore, the "true correlation" between any two metabolites is dependent on the "direct" correlation between them and the "indirect" correlations between any multiset of ancillary metabolites that connect them. To this end, we will define a matrix $M$ as the following:

$$M = \begin{bmatrix} 0 & A \\ A^T & 0 \end{bmatrix} \qquad (1)$$

where $A$ is the adjacency matrix and $0$ is an $n$ x $n$ matrix whose entries are all zero. $M$ is symmetric and it has four sections including (clockwise) matrices that describe the correlations between pairs of metabolites in blood, pairs where the first metabolite is in CSF and the second is in blood, pairs of metabolites in CSF, and pairs where the first metabolite is in blood and the second is in CSF. Because we are only interested in correlations between metabolites across the blood-brain barrier, we set to zero the matrices that describe correlations between pairs of metabolites in the same biofluid. Following this notation the "indirect" correlation between metabolites $j$ in blood and $i$ in blood that goes through a metabolite $k$ in CSF for all $k$ is

calculated as:

$$\sum_{k=1}^{n} A_{ik}A_{jk} = [AA^T]_{ij} \qquad (2)$$

Similarly, the "indirect" correlation between metabolite $j$ in CSF and $i$ in blood through all metabolites in CSF ($k$ = 1 to $n$) and all metabolites ($l$ = 1 to $n$) in blood is:

$$\sum_{k=1}^{n}\sum_{l=1}^{n} A_{ik}A_{lk}A_{ij} = [AA^TA]_{ij} \qquad (3)$$

Then in matrices:

$$M^{2k} = \begin{bmatrix} (AA^T)^k & 0 \\ 0 & (A^TA)^k \end{bmatrix} \text{ and } M^{2k+1} = \begin{bmatrix} 0 & (AA^T)^k A \\ (A^TA)^k A^T & 0 \end{bmatrix} \qquad (4)$$

$j^{th}$ column sum in $M^{2k}$ describes how connected is metabolite $j$ in blood to all metabolites in blood through $2k$-1 intermediary metabolites. The $j^{th}$ column sum of matrix $M^{2k+1}$ describes how connected is metabolite $j$ in blood to all metabolites in CSF through $2k$ intermediary metabolites.

To determine the true level of connection for any given metabolite to all identified metabolites, we must compute column sums of the following series, assuming convergence:

$$T = \sum_{k=1}^{\infty} M^k \qquad (5)$$

For this geometric series, convergence is guaranteed if $|\rho(M)| < 1$ where $\rho(M)$ is the greatest eigenvalue of $M$ in absolute value. Any norm of $M$ acts as an upper bound for all eigenvalues of $M$ so we can scale the matrix by $\alpha$ so that: $|\alpha M| < 1$. Choosing $\alpha > 0$ and using the Frobenius norm as the norm leads to: $\|\alpha M\|_F = |\alpha|\|M\|_F < 1 \leftrightarrow |\alpha| < \|M\|_F^{-1}$ and therefore we can just pick $\alpha = (0.01 + \|M\|_F)^{-1}$ as it satisfies the inequality. With series now converging for $\alpha M$ we can find $T$ (Eq 5). As we are looking for a way to rank the metabolites based on how connected they are to all the others we are interested in the comparison between column sums and not their raw values and therefore after multiplying $M$ with $\alpha$:

$$T = \sum_{k=1}^{\infty} (\alpha M)^k \propto M \sum_{k=1}^{\infty} (\alpha M)^k$$

And

$$T' = M \sum_{k=1}^{\infty} (\alpha M)^k = M(I - \alpha M)^{-1}$$

ranks the connectedness of metabolites based on $T'$ instead of $T$. We measure the connectedness of each metabolite with its column sum in order to rank level of connection for each metabolite. Since the first $n$-columns correspond to metabolites in blood and the last $n$-columns correspond to metabolites in CSF, we rank the metabolites in blood and CSF using the first $n$ and the last $n$ columns respectively.

## Supporting information

**S1 Fig. Average value of metabolites with standard deviation determined in CSF samples determined from 1D NOESY measurements.** Shown are values prior to normalization. (PPTX)

**S2 Fig. Average value of metabolites with standard deviation determined in Blood samples determined from CPMG experiment.** Shown are values prior to normalization.
(PPTX)

## Author Contributions

**Conceptualization:** Kerry Rennie, Balu Chakravarthy, Will J. Costain, Miroslava Cuperlovic-Culf.

**Data curation:** Mariam Taktek, Miroslava Cuperlovic-Culf.

**Formal analysis:** Mariam Taktek, Nam Huan Khieu, Junzhou Huang, Kerry Rennie.

**Funding acquisition:** Miroslava Cuperlovic-Culf.

**Investigation:** Filip Stojanovic, Mariam Taktek, Nam Huan Khieu, Junzhou Huang, Susan Jiang, Miroslava Cuperlovic-Culf.

**Methodology:** Junzhou Huang, Susan Jiang, Kerry Rennie, Will J. Costain, Miroslava Cuperlovic-Culf.

**Project administration:** Will J. Costain, Miroslava Cuperlovic-Culf.

**Resources:** Balu Chakravarthy, Will J. Costain, Miroslava Cuperlovic-Culf.

**Software:** Filip Stojanovic.

**Supervision:** Balu Chakravarthy, Miroslava Cuperlovic-Culf.

**Validation:** Miroslava Cuperlovic-Culf.

**Visualization:** Miroslava Cuperlovic-Culf.

**Writing – original draft:** Miroslava Cuperlovic-Culf.

**Writing – review & editing:** Filip Stojanovic, Mariam Taktek, Nam Huan Khieu, Junzhou Huang, Susan Jiang, Kerry Rennie, Balu Chakravarthy, Will J. Costain, Miroslava Cuperlovic-Culf.

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
