## [Decision Letter · Decision Letter 0]

20 Oct 2020

PONE-D-20-20194

NMR Analysis of the Correlation of Metabolic Changes in Blood and Cerebrospinal Fluid in Alzheimer Model Male and Female Mice

PLOS ONE

Dear Dr. Cuperlovic-Culf,

Thank you for submitting your manuscript to PLOS ONE. After careful consideration by a Reviewer and an Academic Editor, all of the critiques of the Reviewer must be addressed in detail in a revision to determine publication status. If you are prepared to undertake the work required, I would be pleased to reconsider my decision, but revision of the original submission without directly addressing the critiques of the Reviewer does not guarantee acceptance for publication in PLOS ONE. If the authors do not feel that the queries can be addressed, please consider submitting to another publication medium. A revised submission will be sent out for re-review. The authors are urged to have the manuscript given a hard copyedit for syntax and grammar.

**Comments to the Author**

1. Is the manuscript technically sound, and do the data support the conclusions?

Reviewer #1: Yes

2. Has the statistical analysis been performed appropriately and rigorously? 

Reviewer #1: No

3. Have the authors made all data underlying the findings in their manuscript fully available?

Reviewer #1: Yes

4. Is the manuscript presented in an intelligible fashion and written in standard English?

Reviewer #1: Yes

5. Review Comments to the Author

Reviewer #1: The Authors present a nice work in NMR analysis on blood and CSF samples in a Alzheimer mice model. A total of 30 samples (90 mice) were prepared and NMR spectra were recorded using NOESY and CPMG pulse sequences. Data were qualitatively and quantitatively analyzed and statistical analysis were performed. 51 metabolite were identified and correlations between metabolites in blood and CSF were determined.

I have few questions :

- Authors don’t use buffer to prepare the samples and never give the pH of the samples. For all samples the pH are the same ?

- Authors indicate that the region “0.5–9.5 ppm were included in the normalization and analysis”, but residual signal of water is present around 4.7 ppm ? Perhaps the information is in the sentence “…, removal of undesired areas, …’ data analysis section, but give more details please.

- Please give more information about the normalization procedure, it’s not clear.

- How many mice the authors have per group ? If it’s only 3, it’s very poor to give reliable results. I have never seem in this paper, the standard deviation on the results, why ?

minor correction

- table 1, Column Male/blood, Pantothenic acid and not pantothenicacid

6. PLOS authors have the option to publish the peer review history of their article (what does this mean?). If published, this will include your full peer review and any attached files.

**Do you want your identity to be public for this peer review?** For information about this choice, including consent withdrawal, please see our Privacy Policy.

Reviewer #1: No

We look forward to receiving your revised manuscript.

Kind regards,

Stephen D. Ginsberg, Ph.D.

Section Editor

PLOS ONE

3.PLOS requires an ORCID iD for the corresponding author in Editorial Manager on papers submitted after December 6th, 2016. Please ensure that you have an ORCID iD and that it is validated in Editorial Manager. To do this, go to ‘Update my Information’ (in the upper left-hand corner of the main menu), and click on the Fetch/Validate link next to the ORCID field. This will take you to the ORCID site and allow you to create a new iD or authenticate a pre-existing iD in Editorial Manager. Please see the following video for instructions on linking an ORCID iD to your Editorial Manager account: https://www.youtube.com/watch?v=_xcclfuvtxQ

4.Thank you for including your ethics statement:  "All animal procedures were approved by the NRC’s Animal Care Committee and were in compliance with the Canadian Council of Animal Care guidelines."

Please amend your current ethics statement to include the full name of the ethics committee that approved your specific study.

For additional information about PLOS ONE submissions requirements for ethics oversight of animal work, please refer to http://journals.plos.org/plosone/s/submission-guidelines#loc-animal-research  

<h1>** **</h1>

---

## [Author Response · Author response to Decision Letter 0]

24 Mar 2021

We would like to thank Editor and the Reviewer for carefully assessing our manuscript and very useful comments and suggestions. We have made all the requested changes and specific answers are shown below. These changes made our manuscript stronger and we are grateful to reviewer for this. Answers to specific questions are shown in Italic.

Reviewer #1: The Authors present a nice work in NMR analysis on blood and CSF samples in a Alzheimer mice model. A total of 30 samples (90 mice) were prepared and NMR spectra were recorded using NOESY and CPMG pulse sequences. Data were qualitatively and quantitatively analyzed and statistical analysis were performed. 51 metabolite were identified and correlations between metabolites in blood and CSF were determined.

We would like to thank reviewer for this highly encouraging comment.

- Authors don’t use buffer to prepare the samples and never give the pH of the samples. For all samples the pH are the same ?

Following work presented by Canueto et al. (Cañueto, D., Salek, R.M., Correig, X. et al. Improving sample classification by harnessing the potential of 1H-NMR signal chemical shifts. Sci Rep 8, 11886 (2018).) we realized the possibility to utilize chemical shift changes as another metric in assessing the disease effects on body fluid metabolome. Because of that we did not include buffer in the samples. However, our peak position analysis (show through Figure 1) shows now peaks shift that would indicate differences in pH values across samples. Still, in order to correct any possible errors in quantification caused by peak shift we have performed computational peak alignment. 

- Authors indicate that the region “0.5–9.5 ppm were included in the normalization and analysis”, but residual signal of water is present around 4.7 ppm ? Perhaps the information is in the sentence “…, removal of undesired areas, …’ data analysis section, but give more details please.

Water peak region from 4.5-5.2 ppm was removed (set to 0 intensity) from the analysis. We have added this information to the manuscript.

- Please give more information about the normalization procedure, it’s not clear.

Initially all NMR peaks were normalized to the reference peak. Concentrations of metabolites obtained from the spectra were normalized using z-score normalization. This information is added to the manuscript.

- How many mice the authors have per group ? If it’s only 3, it’s very poor to give reliable results. I have never seem in this paper, the standard deviation on the results, why ?

In the experiment we used CSF and blood samples from 90 mice. For CSF we had to pool samples for 3 mice in order to get sufficiently strong signal and we had 3 pools (total of 9 mice in total for each category). For blood we had 3 mice for each category. Although it would be of course very beneficial to have a larger pool of mice for this work at this point this was not possible. Additionally, APP2/PS1 mice have a very short life expectancy making this experiment very difficult as it required a larger pool to insure sufficient survival rate. We have now added as supplementary figures bar graphs for all metabolites showing standard variation across samples. This analysis shows only minor variation. As these animals are all kept at exactly the same conditions and are clonal there are no confounding factors other than differences studied here. 

minor correction

- table 1, Column Male/blood, Pantothenic acid and not pantothenicacid

All metabolite labels have been corrected.

---

## [Decision Letter · Decision Letter 1]

12 Apr 2021

NMR Analysis of the Correlation of Metabolic Changes in Blood and Cerebrospinal Fluid in Alzheimer Model Male and Female Mice

PONE-D-20-20194R1

Dear Dr. Cuperlovic-Culf,

We’re pleased to inform you that your manuscript has been judged scientifically suitable for publication and will be formally accepted for publication once it meets all outstanding technical requirements.

Kind regards,

Stephen D. Ginsberg, Ph.D.

Section Editor

PLOS ONE

**Comments to the Author**

1. If the authors have adequately addressed your comments raised in a previous round of review and you feel that this manuscript is now acceptable for publication, you may indicate that here to bypass the “Comments to the Author” section, enter your conflict of interest statement in the “Confidential to Editor” section, and submit your "Accept" recommendation.

Reviewer #1: All comments have been addressed

2. Is the manuscript technically sound, and do the data support the conclusions?

Reviewer #1: Yes

3. Has the statistical analysis been performed appropriately and rigorously? 

Reviewer #1: Yes

4. Have the authors made all data underlying the findings in their manuscript fully available?

Reviewer #1: Yes

5. Is the manuscript presented in an intelligible fashion and written in standard English?

Reviewer #1: Yes

6. Review Comments to the Author

Reviewer #1: I thank the authors for their precision. This article can be accpet and publish in the PLOSONE journal.

7. PLOS authors have the option to publish the peer review history of their article (what does this mean?). If published, this will include your full peer review and any attached files.

Reviewer #1: No

---

## [Editor Report · Acceptance letter]

22 Apr 2021

PONE-D-20-20194R1 

NMR Analysis of the Correlation of Metabolic Changes in Blood and Cerebrospinal Fluid in Alzheimer Model Male and Female Mice 

Dear Dr. Cuperlovic-Culf:

I'm pleased to inform you that your manuscript has been deemed suitable for publication in PLOS ONE. Congratulations! Your manuscript is now with our production department. 

Kind regards, 

on behalf of

Dr. Stephen D. Ginsberg 

Section Editor

PLOS ONE